# Temporal Ensembling for Semi-Supervised Learning

**Samuli Laine**
NVIDIA
slaine@nvidia.com

**Timo Aila**
NVIDIA
taila@nvidia.com

## Abstract

In this paper, we present a simple and efficient method for training deep neural networks in a semi-supervised setting where only a small portion of training data is labeled. We introduce self-ensembling, where we form a consensus prediction of the unknown labels using the outputs of the network-in-training on different epochs, and most importantly, under different regularization and input augmentation conditions. This ensemble prediction can be expected to be a better predictor for the unknown labels than the output of the network at the most recent training epoch, and can thus be used as a target for training. Using our method, we set new records for two standard semi-supervised learning benchmarks, reducing the (non-augmented) classification error rate from 18.44% to 7.05% in SVHN with 500 labels and from 18.63% to 16.55% in CIFAR-10 with 4000 labels, and further to 5.12% and 12.16% by enabling the standard augmentations. We additionally obtain a clear improvement in CIFAR-100 classification accuracy by using random images from the Tiny Images dataset as unlabeled extra inputs during training. Finally, we demonstrate good tolerance to incorrect labels.

## 1 Introduction

It has long been known that an ensemble of multiple neural networks generally yields better predictions than a single network in the ensemble. This effect has also been indirectly exploited when training a single network through dropout (Srivastava et al., 2014), dropconnect (Wan et al., 2013), or stochastic depth (Huang et al., 2016) regularization methods, and in swapout networks (Singh et al., 2016), where training always focuses on a particular subset of the network, and thus the complete network can be seen as an implicit ensemble of such trained sub-networks. We extend this idea by forming ensemble predictions during training, using the outputs of a single network on different training epochs and under different regularization and input augmentation conditions. Our training still operates on a single network, but the predictions made on different epochs correspond to an ensemble prediction of a large number of individual sub-networks because of dropout regularization.

This ensemble prediction can be exploited for semi-supervised learning where only a small portion of training data is labeled. If we compare the ensemble prediction to the current output of the network being trained, the ensemble prediction is likely to be closer to the correct, unknown labels of the unlabeled inputs. Therefore the labels inferred this way can be used as training targets for the unlabeled inputs. Our method relies heavily on dropout regularization and versatile input augmentation. Indeed, without neither, there would be much less reason to place confidence in whatever labels are inferred for the unlabeled training data.

We describe two ways to implement self-ensembling, $\Pi$-model and temporal ensembling. Both approaches surpass prior state-of-the-art results in semi-supervised learning by a considerable margin. We furthermore observe that self-ensembling improves the classification accuracy in fully labeled cases as well, and provides tolerance against incorrect labels.

The recently introduced transform/stability loss of Sajjadi et al. (2016b) is based on the same principle as our work, and the $\Pi$-model can be seen as a special case of it. The $\Pi$-model can also be seen as a simplification of the $\Gamma$-model of the ladder network by Rasmus et al. (2015), a previously presented network architecture for semi-supervised learning. Our temporal ensembling method has connections to the bootstrapping method of Reed et al. (2014) targeted for training with noisy labels.

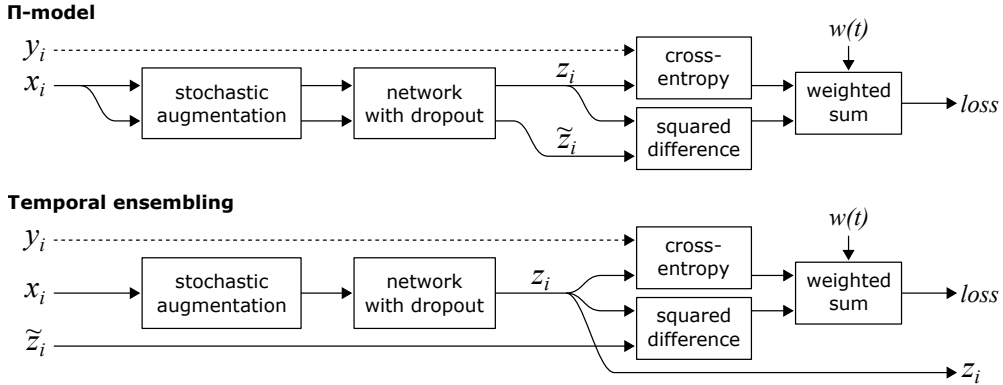

Figure 1: Structure of the training pass in our methods. Top: $\Pi$-model. Bottom: temporal ensembling. Labels $y_i$ are available only for the labeled inputs, and the associated cross-entropy loss component is evaluated only for those.

---

**Algorithm 1** $\Pi$-model pseudocode.

---

**Require:** $x_i$ = training stimuli
**Require:** $L$ = set of training input indices with known labels
**Require:** $y_i$ = labels for labeled inputs $i \in L$
**Require:** $w(t)$ = unsupervised weight ramp-up function
**Require:** $f_\theta(x)$ = stochastic neural network with trainable parameters $\theta$
**Require:** $g(x)$ = stochastic input augmentation function
 **for** $t$ in $[1, num\_epochs]$ **do**
 **for** each minibatch $B$ **do**
 $z_{i \in B} \leftarrow f_\theta(g(x_{i \in B}))$ $\triangleright$ evaluate network outputs for augmented inputs
 $\tilde{z}_{i \in B} \leftarrow f_\theta(g(x_{i \in B}))$ $\triangleright$ again, with different dropout and augmentation
 $loss \leftarrow -\frac{1}{|B|} \sum_{i \in (B \cap L)} \log z_i[y_i]$ $\triangleright$ supervised loss component
 $+ w(t)\frac{1}{C|B|} \sum_{i \in B} ||z_i - \tilde{z}_i||^2$ $\triangleright$ unsupervised loss component
 update $\theta$ using, e.g., ADAM $\triangleright$ update network parameters
 **end for**
 **end for**
 **return** $\theta$

---

## 2 Self-ensembling during training

We present two implementations of self-ensembling during training. The first one, $\Pi$-model, encourages consistent network output between two realizations of the same input stimulus, under two different dropout conditions. The second method, temporal ensembling, simplifies and extends this by taking into account the network predictions over multiple previous training epochs.

We shall describe our methods in the context of traditional image classification networks. Let the training data consist of total of $N$ inputs, out of which $M$ are labeled. The input stimuli, available for all training data, are denoted $x_i$, where $i \in \{1 \ldots N\}$. Let set $L$ contain the indices of the labeled inputs, $|L| = M$. For every $i \in L$, we have a known correct label $y_i \in \{1 \ldots C\}$, where $C$ is the number of different classes.

### 2.1 $\Pi$-model

The structure of $\Pi$-model is shown in Figure 1 (top), and the pseudocode in Algorithm 1. During training, we evaluate the network for each training input $x_i$ twice, resulting in prediction vectors $z_i$ and $\tilde{z}_i$. Our loss function consists of two components. The first component is the standard cross-entropy loss, evaluated for labeled inputs only. The second component, evaluated for all inputs, penalizes different predictions for the same training input $x_i$ by taking the mean square difference

between the prediction vectors $z_i$ and $\tilde{z}_i$.[1] To combine the supervised and unsupervised loss terms, we scale the latter by time-dependent weighting function $w(t)$. By comparing the entire output vectors $z_i$ and $\tilde{z}_i$, we effectively ask the "dark knowledge" (Hinton et al., 2015) between the two evaluations to be close, which is a much stronger requirement compared to asking that only the final classification remains the same, which is what happens in traditional training.

It is important to notice that, because of dropout regularization, the network output during training is a stochastic variable. Thus two evaluations of the same input $x_i$ under same network weights $\theta$ yield different results. In addition, Gaussian noise and augmentations such as random translation are evaluated twice, resulting in additional variation. The combination of these effects explains the difference between the prediction vectors $z_i$ and $\tilde{z}_i$. This difference can be seen as an error in classification, given that the original input $x_i$ was the same, and thus minimizing it is a reasonable goal.

In our implementation, the unsupervised loss weighting function $w(t)$ ramps up, starting from zero, along a Gaussian curve during the first 80 training epochs. See Appendix A for further details about this and other training parameters. In the beginning the total loss and the learning gradients are thus dominated by the supervised loss component, i.e., the labeled data only. We have found it to be very important that the ramp-up of the unsupervised loss component is slow enough—otherwise, the network gets easily stuck in a degenerate solution where no meaningful classification of the data is obtained.

Our approach is somewhat similar to the $\Gamma$-model of the ladder network by Rasmus et al. (2015), but conceptually simpler. In the $\Pi$-model, the comparison is done directly on network outputs, i.e., after softmax activation, and there is no auxiliary mapping between the two branches such as the learned denoising functions in the ladder network architecture. Furthermore, instead of having one "clean" and one "corrupted" branch as in $\Gamma$-model, we apply equal augmentation and noise to the inputs for both branches.

As shown in Section 3, the $\Pi$-model combined with a good convolutional network architecture provides a significant improvement over prior art in classification accuracy.

## 2.2 TEMPORAL ENSEMBLING

Analyzing how the $\Pi$-model works, we could equally well split the evaluation of the two branches in two separate phases: first classifying the training set once without updating the weights $\theta$, and then training the network on the same inputs under different augmentations and dropout, using the just obtained predictions as targets for the unsupervised loss component. As the training targets obtained this way are based on a single evaluation of the network, they can be expected to be noisy. Temporal ensembling alleviates this by aggregating the predictions of multiple previous network evaluations into an ensemble prediction. It also lets us evaluate the network only once during training, gaining an approximate 2x speedup over the $\Pi$-model.

The structure of our temporal ensembling method is shown in Figure 1 (bottom), and the pseudocode in Algorithm 2. The main difference to the $\Pi$-model is that the network and augmentations are evaluated only once per input per epoch, and the target vectors $\tilde{z}$ for the unsupervised loss component are based on prior network evaluations instead of a second evaluation of the network.

After every training epoch, the network outputs $z_i$ are accumulated into ensemble outputs $Z_i$ by updating $Z_i \leftarrow \alpha Z_i + (1 - \alpha)z_i$, where $\alpha$ is a momentum term that controls how far the ensemble reaches into training history. Because of dropout regularization and stochastic augmentation, $Z$ thus contains a weighted average of the outputs of an ensemble of networks $f$ from previous training epochs, with recent epochs having larger weight than distant epochs. For generating the training targets $\tilde{z}$, we need to correct for the startup bias in $Z$ by dividing by factor $(1 - \alpha^t)$. A similar bias correction has been used in, e.g., Adam (Kingma & Ba, 2014) and mean-only batch normalization (Salimans & Kingma, 2016). On the first training epoch, $Z$ and $\tilde{z}$ are zero as no data from previous epochs is available. For this reason, we specify the unsupervised weight ramp-up function $w(t)$ to also be zero on the first training epoch.

---

[1] Squared difference gave slightly but consistently better results than cross-entropy loss in our tests.

---

**Algorithm 2** Temporal ensembling pseudocode. Note that the updates of $Z$ and $\tilde{z}$ could equally well be done inside the minibatch loop; in this pseudocode they occur between epochs for clarity.

---

**Require:** $x_i$ = training stimuli
**Require:** $L$ = set of training input indices with known labels
**Require:** $y_i$ = labels for labeled inputs $i \in L$
**Require:** $\alpha$ = ensembling momentum, $0 \leq \alpha < 1$
**Require:** $w(t)$ = unsupervised weight ramp-up function
**Require:** $f_\theta(x)$ = stochastic neural network with trainable parameters $\theta$
**Require:** $g(x)$ = stochastic input augmentation function
 $Z \leftarrow \mathbf{0}_{[N \times C]}$ ▷ initialize ensemble predictions
 $\tilde{z} \leftarrow \mathbf{0}_{[N \times C]}$ ▷ initialize target vectors
 **for** $t$ in $[1, num\_epochs]$ **do**
 **for** each minibatch $B$ **do**
 $z_{i \in B} \leftarrow f_\theta(g(x_{i \in B}, t))$ ▷ evaluate network outputs for augmented inputs
 $loss \leftarrow -\frac{1}{|B|} \sum_{i \in (B \cap L)} \log z_i[y_i]$ ▷ supervised loss component
 $+ w(t)\frac{1}{C|B|} \sum_{i \in B} ||z_i - \tilde{z}_i||^2$ ▷ unsupervised loss component
 update $\theta$ using, e.g., ADAM ▷ update network parameters
 **end for**
 $Z \leftarrow \alpha Z + (1 - \alpha)z$ ▷ accumulate ensemble predictions
 $\tilde{z} \leftarrow Z/(1 - \alpha^t)$ ▷ construct target vectors by bias correction
 **end for**
 **return** $\theta$

---

The benefits of temporal ensembling compared to $\Pi$-model are twofold. First, the training is faster because the network is evaluated only once per input on each epoch. Second, the training targets $\tilde{z}$ can be expected to be less noisy than with $\Pi$-model. As shown in Section 3, we indeed obtain somewhat better results with temporal ensembling than with $\Pi$-model in the same number of training epochs. The downside compared to $\Pi$-model is the need to store auxiliary data across epochs, and the new hyperparameter $\alpha$. While the matrix $Z$ can be fairly large when the dataset contains a large number of items and categories, its elements are accessed relatively infrequently. Thus it can be stored, e.g., in a memory mapped file.

An intriguing additional possibility of temporal ensembling is collecting other statistics from the network predictions $z_i$ besides the mean. For example, by tracking the second raw moment of the network outputs, we can estimate the variance of each output component $z_{i,j}$. This makes it possible to reason about the uncertainty of network outputs in a principled way (Gal & Ghahramani, 2016). Based on this information, we could, e.g., place more weight on more certain predictions vs. uncertain ones in the unsupervised loss term. However, we leave the exploration of these avenues as future work.

## 3 RESULTS

Our network structure is given in Table 5, and the test setup and all training parameters are detailed in Appendix A. We test the $\Pi$-model and temporal ensembling in two image classification tasks, CIFAR-10 and SVHN, and report the mean and standard deviation of 10 runs using different random seeds.

Although it is rarely stated explicitly, we believe that our comparison methods do not use input augmentation, i.e., are limited to dropout and other forms of permutation-invariant noise. Therefore we report the error rates without augmentation, unless explicitly stated otherwise. Given that the ability of an algorithm to extract benefit from augmentation is also an important property, we report the classification accuracy using a standard set of augmentations as well. In purely supervised training the de facto standard way of augmenting the CIFAR-10 dataset includes horizontal flips and random translations, while SVHN is limited to random translations. By using these same augmentations we can compare against the best fully supervised results as well. After all, the fully supervised results should indicate the upper bound of obtainable accuracy.

Table 1: CIFAR-10 results with 4000 labels, averages of 10 runs (4 runs for all labels).

|  | Error rate (%) with # labels | |
| --- | --- | --- |
|  | 4000 | All (50000) |
| Supervised-only | $35.56 \pm 1.59$ | $7.33 \pm 0.04$ |
| with augmentation | $34.85 \pm 1.65$ | $6.05 \pm 0.15$ |
| Conv-Large, $\Gamma$-model (Rasmus et al., 2015) | $20.40 \pm 0.47$ | |
| CatGAN (Springenberg, 2016) | $19.58 \pm 0.58$ | |
| GAN of Salimans et al. (2016) | $18.63 \pm 2.32$ | |
| $\Pi$-model | $16.55 \pm 0.29$ | $6.90 \pm 0.07$ |
| $\Pi$-model with augmentation | $12.36 \pm 0.31$ | $\mathbf{5.56 \pm 0.10}$ |
| Temporal ensembling with augmentation | $\mathbf{12.16 \pm 0.24}$ | $5.60 \pm 0.10$ |

Table 2: SVHN results for 500 and 1000 labels, averages of 10 runs (4 runs for all labels).

| Model | Error rate (%) with # labels | | |
| --- | --- | --- | --- |
|  | 500 | 1000 | All (73257) |
| Supervised-only | $35.18 \pm 5.61$ | $20.47 \pm 2.64$ | $3.05 \pm 0.07$ |
| with augmentation | $31.59 \pm 3.60$ | $19.30 \pm 3.89$ | $2.88 \pm 0.03$ |
| DGN (Kingma et al., 2014) | | $36.02 \pm 0.10$ | |
| Virtual Adversarial (Miyato et al., 2016) | | $24.63$ | |
| ADGM (Maaløe et al., 2016) | | $22.86$ | |
| SDGM (Maaløe et al., 2016) | | $16.61 \pm 0.24$ | |
| GAN of Salimans et al. (2016) | $18.44 \pm 4.8$ | $8.11 \pm 1.3$ | |
| $\Pi$-model | $7.05 \pm 0.30$ | $5.43 \pm 0.25$ | $2.78 \pm 0.03$ |
| $\Pi$-model with augmentation | $6.65 \pm 0.53$ | $4.82 \pm 0.17$ | $\mathbf{2.54 \pm 0.04}$ |
| Temporal ensembling with augmentation | $\mathbf{5.12 \pm 0.13}$ | $\mathbf{4.42 \pm 0.16}$ | $2.74 \pm 0.06$ |

## 3.1 CIFAR-10

CIFAR-10 is a dataset consisting of $32 \times 32$ pixel RGB images from ten classes. Table 1 shows a 2.1 percentage point reduction in classification error rate with 4000 labels (400 per class) compared to earlier methods for the non-augmented $\Pi$-model.

Enabling the standard set of augmentations further reduces the error rate by 4.2 percentage points to 12.36%. Temporal ensembling is slightly better still at 12.16%, while being twice as fast to train. This small improvement conceals the subtle fact that random horizontal flips need to be done independently for each epoch in temporal ensembling, while $\Pi$-model can randomize once per a pair of evaluations, which according to our measurements is $\sim 0.5$ percentage points better than independent flips.

A principled comparison with Sajjadi et al. (2016b) is difficult due to several reasons. They provide results only for a fairly extreme set of augmentations (translations, flipping, rotations, stretching, and shearing) on top of fractional max pooling (Graham, 2014), which introduces random, local stretching inside the network, and is known to improve classification results substantially. They quote an error rate of only 13.60% for supervised-only training with 4000 labels, while our corresponding baseline is 34.85%. This gap indicates a huge benefit from versatile augmentations and fractional max pooling—in fact, their baseline result is already better than any previous semi-supervised results. By enabling semi-supervised learning they achieve a 17% drop in classification error rate (from 13.60% to 11.29%), while we see a much larger relative drop of 65% (from 34.85% to 12.16%).

## 3.2 SVHN

The street view house numbers (SVHN) dataset consists of $32 \times 32$ pixel RGB images of real-world house numbers, and the task is to classify the centermost digit. In SVHN we chose to use only the

Table 3: CIFAR-100 results with 10000 labels, averages of 10 runs (4 runs for all labels).

| | Error rate (%) with # labels | |
|---|---|---|
| | 10000 | All (50000) |
| Supervised-only | $51.21 \pm 0.33$ | $29.14 \pm 0.25$ |
| with augmentation | $44.56 \pm 0.30$ | $26.42 \pm 0.17$ |
| $\Pi$-model | $43.43 \pm 0.54$ | $29.06 \pm 0.21$ |
| $\Pi$-model with augmentation | $39.19 \pm 0.36$ | $26.32 \pm 0.04$ |
| Temporal ensembling with augmentation | $\mathbf{38.65 \pm 0.51}$ | $\mathbf{26.30 \pm 0.15}$ |

Table 4: CIFAR-100 + Tiny Images results, averages of 10 runs.

| | Error rate (%) with # unlabeled auxiliary inputs from Tiny Images | |
|---|---|---|
| | Random 500k | Restricted 237k |
| $\Pi$-model with augmentation | $25.79 \pm 0.17$ | $25.43 \pm 0.32$ |
| Temporal ensembling with augmentation | $\mathbf{23.62 \pm 0.23}$ | $\mathbf{23.79 \pm 0.24}$ |

official 73257 training examples following Salimans et al. (2016). Even with this choice our error rate with all labels is only $3.05\%$ without augmentation.

Table 2 compares our method to the previous state-of-the-art. With the most commonly used 1000 labels we observe an improvement of 2.7 percentage points, from $8.11\%$ to $5.43\%$ without augmentation, and further to $4.42\%$ with standard augmentations.

We also investigated the behavior with 500 labels, where we obtained an error rate less than half of Salimans et al. (2016) without augmentations, with a significantly lower standard deviation as well. When augmentations were enabled, temporal ensembling further reduced the error rate to $5.12\%$. In this test the difference between $\Pi$-model and temporal ensembling was quite significant at 1.5 percentage points.

In SVHN Sajjadi et al. (2016b) provide results without augmentation, with the caveat that they use fractional max pooling, which is a very augmentation-like technique due to the random, local stretching it introduces inside the network. It leads to a superb error rate of $2.28\%$ in supervised-only training, while our corresponding baseline is $3.05\%$ (or $2.88\%$ with translations). Given that in a separate experiment our network matched the best published result for non-augmented SVHN when extra data is used ($1.69\%$ from Lee et al. (2015)), this gap is quite surprising, and leads us to conclude that fractional max pooling leads to a powerful augmentation of the dataset, well beyond what simple translations can achieve. Our temporal ensembling technique obtains better error rates for both 500 and 1000 labels ($5.12\%$ and $4.42\%$, respectively) compared to the $6.03\%$ reported by Sajjadi et al. for 732 labels.

### 3.3 CIFAR-100 AND TINY IMAGES

The CIFAR-100 dataset consists of $32 \times 32$ pixel RGB images from a hundred classes. We are not aware of previous semi-supervised results in this dataset, and chose 10000 labels for our experiments. Table 3 shows error rates of $43.43\%$ and $38.65\%$ without and with augmentation, respectively. These correspond to 7.8 and 5.9 percentage point improvements compared to supervised learning with labeled inputs only.

We ran two additional tests using unlabeled extra data from Tiny Images dataset (Torralba et al., 2008): one with randomly selected 500k extra images, most not corresponding to any of the CIFAR-100 categories, and another with a restricted set of 237k images from the categories that correspond to those found in the CIFAR-100 dataset (see appendix A for details). The results are shown in Table 4. The addition of randomly selected, unlabeled extra images improved the error rate by 2.7 percentage points (from $26.30\%$ to $23.63\%$), indicating a desirable ability to learn from random natural images. Temporal ensembling benefited much more from the extra data than the $\Pi$-model. Interestingly, restricting the extra data to categories that are present in CIFAR-100 did not improve

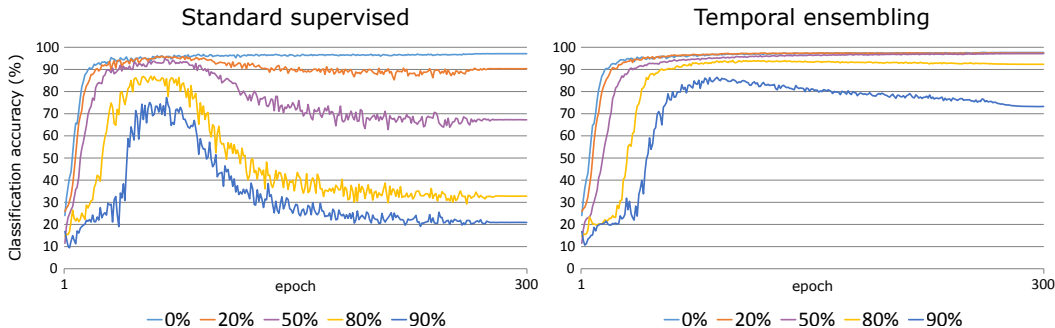

Figure 2: Percentage of correct SVHN classifications as a function of training epoch when a part of the labels is randomized. With standard supervised training (left) the classification accuracy suffers when even a small portion of the labels give disinformation, and the situation worsens quickly as the portion of randomized labels increases to 50% or more. On the other hand, temporal ensembling (right) shows almost perfect resistance to disinformation when half of the labels are random, and retains over ninety percent classification accuracy even when 80% of the labels are random.

the classification accuracy further. This indicates that in order to train a better classifier by adding extra data as unlabeled inputs, it is enough to have the extra data roughly in the same space as the actual inputs—in our case, natural images. We hypothesize that it may even be possible to use properly crafted synthetic data as unlabeled inputs to obtain improved classifiers.

In order to keep the training times tolerable, we limited the number of unlabeled inputs to 50k per epoch in these tests, i.e., on every epoch we trained using all 50k labeled inputs from CIFAR-100 and 50k additional unlabeled inputs from Tiny Images. The 50k unlabeled inputs were chosen randomly on each epoch from the 500k or 237k extra inputs. In temporal ensembling, after each epoch we updated only the rows of $Z$ that corresponded to inputs used on that epoch.

### 3.4 SUPERVISED LEARNING

When all labels are used for traditional supervised training, our network approximately matches the state-of-the-art error rate for a single model in CIFAR-10 with augmentation (Lee et al., 2015; Mishkin & Matas, 2016) at $6.05\%$, and without augmentation (Salimans & Kingma, 2016) at $7.33\%$. The same is probably true for SVHN as well, but there the best published results rely on extra data that we chose not to use.

Given this premise, it is perhaps somewhat surprising that our methods reduce the error rate also when all labels are used (Tables 1 and 2). We believe that this is an indication that the consistency requirement adds a degree of resistance to ambiguous labels that are fairly common in many classification tasks, and that it encourages features to be more invariant to stochastic sampling.

### 3.5 TOLERANCE TO INCORRECT LABELS

In a further test we studied the hypothesis that our methods add tolerance to incorrect labels by assigning a random label to a certain percentage of the training set before starting to train. Figure 2 shows the classification error graphs for standard supervised training and temporal ensembling.

Clearly our methods provide considerable resistance to wrong labels, and we believe this is because the unsupervised loss term encourages the mapping function implemented by the network to be flat in the vicinity of all input data points, whereas the supervised loss term enforces the mapping function to have a specific value in the vicinity of the labeled input data points. This means that even the wrongly labeled inputs play a role in shaping the mapping function—the unsupervised loss term smooths the mapping function and thus also the decision boundaries, effectively fusing the inputs into coherent clusters, whereas the excess of correct labels in each class is sufficient for locking the clusters to the right output vectors through the supervised loss term. The difference to classical regularizers is that we induce smoothness only on the manifold of likely inputs instead

of over the entire input domain. For further analysis about the importance of the gradient of the mapping function, see Simard et al. (1998).

## 4 RELATED WORK

There is a large body of previous work on semi-supervised learning (Zhu, 2005). In here we will concentrate on the ones that are most directly connected to our work.

$\Gamma$-model is a subset of a ladder network (Rasmus et al., 2015) that introduces lateral connections into an encoder-decoder type network architecture, targeted at semi-supervised learning. In $\Gamma$-model, all but the highest lateral connections in the ladder network are removed, and after pruning the unnecessary stages, the remaining network consists of two parallel, identical branches. One of the branches takes the original training inputs, whereas the other branch is given the same input corrupted with noise. The unsupervised loss term is computed as the squared difference between the (pre-activation) output of the clean branch and a denoised (pre-activation) output of the corrupted branch. The denoised estimate is computed from the output of the corrupted branch using a parametric nonlinearity that has 10 auxiliary trainable parameters per unit. Our $\Pi$-model differs from the $\Gamma$-model in removing the parametric nonlinearity and denoising, having two corrupted paths, and comparing the outputs of the network instead of pre-activation data of the final layer.

Sajjadi et al. (2016b) recently introduced a new loss function for semi-supervised learning, so called transform/stability loss, which is founded on the same principle as our work. During training, they run augmentation and network evaluation $n$ times for each minibatch, and then compute an unsupervised loss term as the sum of all pairwise squared distances between the obtained $n$ network outputs. As such, their technique follows the general pseudo-ensemble agreement (PEA) regularization framework of Bachman et al. (2014). In addition, they employ a mutual exclusivity loss term (Sajjadi et al., 2016a) that we do not use. Our $\Pi$-model can be seen as a special case of the transform/stability loss obtained by setting $n = 2$. The computational cost of training with transform/stability loss increases linearly as a function of $n$, whereas the efficiency of our temporal ensembling technique remains constant regardless of how large effective ensemble we obtain via the averaging of previous epochs' predictions.

In bootstrap aggregating, or *bagging*, multiple networks are trained independently based on subsets of training data (Breiman, 1996). This results in an ensemble that is more stable and accurate than the individual networks. Our approach can be seen as pulling the predictions from an implicit ensemble that is based on a single network, and the variability is a result of evaluating it under different dropout and augmentation conditions instead of training on different subsets of data. In work parallel to ours, Huang et al. (2017) store multiple snapshots of the network during training, hopefully corresponding to different local minima, and use them as an explicit ensemble.

The general technique of inferring new labels from partially labeled data is often referred to as *bootstrapping* or *self-training*, and it was first proposed by Yarowsky (1995) in the context of linguistic analysis. Whitney & Sarkar (2012) analyze Yarowsky's algorithm and propose a novel graph-based label propagation approach. Similarly, label propagation methods (Zhu & Ghahramani, 2002) infer labels for unlabeled training data by comparing the associated inputs to labeled training inputs using a suitable distance metric. Our approach differs from this in two important ways. Firstly, we never compare training inputs against each other, but instead only rely on the unknown labels remaining constant, and secondly, we let the network produce the likely classifications for the unlabeled inputs instead of providing them through an outside process.

In addition to partially labeled data, considerable amount of effort has been put into dealing with densely but inaccurately labeled data. This can be seen as a semi-supervised learning task where part of the training process is to identify the labels that are not to be trusted. For recent work in this area, see, e.g., Sukhbaatar et al. (2014) and Patrini et al. (2016). In this context of noisy labels, Reed et al. (2014) presented a simple bootstrapping method that trains a classifier with the target composed of a convex combination of the previous epoch output and the known but potentially noisy labels. Our temporal ensembling differs from this by taking into account the evaluations over multiple epochs.

Generative Adversarial Networks (GAN) have been recently used for semi-supervised learning with promising results (Maaløe et al., 2016; Springenberg, 2016; Odena, 2016; Salimans et al., 2016). It

Table 5: The network architecture used in all of our tests.

| NAME | DESCRIPTION |
|---|---|
| input | $32 \times 32$ RGB image |
| noise | Additive Gaussian noise $\sigma = 0.15$ |
| conv1a | 128 filters, $3 \times 3$, pad = 'same', LReLU ($\alpha = 0.1$) |
| conv1b | 128 filters, $3 \times 3$, pad = 'same', LReLU ($\alpha = 0.1$) |
| conv1c | 128 filters, $3 \times 3$, pad = 'same', LReLU ($\alpha = 0.1$) |
| pool1 | Maxpool $2 \times 2$ pixels |
| drop1 | Dropout, $p = 0.5$ |
| conv2a | 256 filters, $3 \times 3$, pad = 'same', LReLU ($\alpha = 0.1$) |
| conv2b | 256 filters, $3 \times 3$, pad = 'same', LReLU ($\alpha = 0.1$) |
| conv2c | 256 filters, $3 \times 3$, pad = 'same', LReLU ($\alpha = 0.1$) |
| pool2 | Maxpool $2 \times 2$ pixels |
| drop2 | Dropout, $p = 0.5$ |
| conv3a | 512 filters, $3 \times 3$, pad = 'valid', LReLU ($\alpha = 0.1$) |
| conv3b | 256 filters, $1 \times 1$, LReLU ($\alpha = 0.1$) |
| conv3c | 128 filters, $1 \times 1$, LReLU ($\alpha = 0.1$) |
| pool3 | Global average pool ($6 \times 6 \to 1 \times 1$ pixels) |
| dense | Fully connected $128 \to 10$ |
| output | Softmax |

could be an interesting avenue for future work to incorporate a generative component to our solution. We also envision that our methods could be applied to regression-type learning tasks.

## 5 ACKNOWLEDGEMENTS

We thank the anonymous reviewers, Tero Karras, Pekka Jänis, Tim Salimans, Ian Goodfellow, as well as Harri Valpola and his colleagues at Curious AI for valuable suggestions that helped to improve this article.

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

## A  NETWORK ARCHITECTURE, TEST SETUP, AND TRAINING PARAMETERS

Table 5 details the network architecture used in all of our tests. It is heavily inspired by ConvPool-CNN-C (Springenberg et al., 2014) and the improvements made by Salimans & Kingma (2016). All data layers were initialized following He et al. (2015), and we applied weight normalization and mean-only batch normalization (Salimans & Kingma, 2016) with momentum 0.999 to all of them. We used leaky ReLU (Maas et al., 2013) with $\alpha = 0.1$ as the non-linearity, and chose to use max pooling instead of strided convolutions because it gave consistently better results in our experiments.

All networks were trained using Adam (Kingma & Ba, 2014) with a maximum learning rate of $\lambda_{max} = 0.003$, except for temporal ensembling in the SVHN case where a maximum learning rate of $\lambda_{max} = 0.001$ worked better. Adam momentum parameters were set to $\beta_1 = 0.9$ and $\beta_2 = 0.999$ as suggested in the paper. The maximum value for the unsupervised loss component was set to $w_{max} \cdot M/N$, where $M$ is the number of labeled inputs and $N$ is the total number of training inputs. For $\Pi$-model runs, we used $w_{max} = 100$ in all runs except for CIFAR-100 with Tiny Images where we set $w_{max} = 300$. For temporal ensembling we used $w_{max} = 30$ in most runs. For the corrupted label test in Section 3.5 we used $w_{max} = 300$ for 0% and 20% corruption, and $w_{max} = 3000$ for corruption of 50% and higher. For basic CIFAR-100 runs we used $w_{max} = 100$, and for CIFAR-100 with Tiny Images we used $w_{max} = 1000$. The accumulation decay constant of temporal ensembling was set to $\alpha = 0.6$ in all runs.

In all runs we ramped up both the learning rate $\lambda$ and unsupervised loss component weight $w$ during the first 80 epochs using a Gaussian ramp-up curve $\exp[-5(1 - T)^2]$, where $T$ advances linearly from zero to one during the ramp-up period. In addition to ramp-up, we annealed the learning rate $\lambda$ to zero and Adam $\beta_1$ to 0.5 during the last 50 epochs, but otherwise we did not decay them during training. The ramp-down curve was similar to the ramp-up curve but time-reversed and with a scaling constant of 12.5 instead of 5. All networks were trained for 300 epochs with minibatch size of 100.

**CIFAR-10**  Following previous work in fully supervised learning, we pre-processed the images using ZCA and augmented the dataset using horizontal flips and random translations. The translations were drawn from $[-2, 2]$ pixels, and were independently applied to both branches in the $\Pi$-model.

**SVHN**  We pre-processed the input images by biasing and scaling each input image to zero mean and unit variance. We used only the 73257 items in the official training set, i.e., did not use the provided 531131 extra items. The training setups were otherwise similar to CIFAR-10 except that horizontal flips were not used.

**Implementation**  Our implementation is written in Python using Theano (Theano Development Team, 2016) and Lasagne (Dieleman et al., 2015), and is available at `https://github.com/smlaine2/tempens`.

**Model convergence**  As discussed in Section 2.1, a slow ramp-up of the unsupervised cost is very important for getting the models to converge. Furthermore, in our very preliminary tests with 250 labels in SVHN we noticed that optimization tended to explode during the ramp-up period, and we eventually found that using a lower value for Adam $\beta_2$ parameter (e.g., 0.99 instead of 0.999) seems to help in this regard.

We do not attempt to guarantee that the occurrence of labeled inputs during training would be somehow stratified; with bad luck there might be several consecutive minibatches without any labeled inputs when the label density is very low. Some previous work has identified this as a weakness, and have solved the issue by shuffling the input sequences in such a way that stratification is guaranteed, e.g. Rasmus et al. (2015) (confirmed from the authors). This kind of stratification might further improve the convergence of our methods as well.

**Tiny Images, extra data from restricted categories**  The restricted extra data in Section 3.3 was extracted from Tiny Images by picking all images with labels corresponding to the 100 categories used in CIFAR-100. As the Tiny Images dataset does not contain CIFAR-100 categories *aquarium_fish* and *maple_tree*, we used images with labels *fish* and *maple* instead. The result was a total of 237 203 images that were used as unlabeled extra data. Table 6 shows the composition of this extra data set.

It is worth noting that the CIFAR-100 dataset itself is a subset of Tiny Images, and we did not explicitly prevent overlap between this extra set and CIFAR-100. This led to approximately a third of the CIFAR-100 training and test images being present as unlabeled inputs in the extra set. The other test with 500k extra entries picked randomly out of all 79 million images had a negligible overlap with CIFAR-100.

Table 6: The Tiny Images (Torralba et al., 2008) labels and image counts used in the CIFAR-100 plus restricted extra data tests (rightmost column of Table 4). Note that the extra input images were supplied as unlabeled data for our networks, and the labels were used only for narrowing down the full set of 79 million images.

| Label | # | Label | # | Label | # | Label | # |
|---|---|---|---|---|---|---|---|
| apple | 2242 | baby | 2771 | bear | 2242 | beaver | 2116 |
| bed | 2767 | bee | 2193 | beetle | 2173 | bicycle | 2599 |
| bottle | 2212 | bowl | 2707 | boy | 2234 | bridge | 2274 |
| bus | 3068 | butterfly | 3036 | camel | 2121 | can | 2461 |
| castle | 3094 | caterpillar | 2382 | cattle | 2089 | chair | 2552 |
| chimpanzee | 1706 | clock | 2375 | cloud | 2390 | cockroach | 2318 |
| couch | 2171 | crab | 2735 | crocodile | 2712 | cup | 2287 |
| dinosaur | 2045 | dolphin | 2504 | elephant | 2794 | fish* | 3082 |
| flatfish | 1504 | forest | 2244 | fox | 2684 | girl | 2204 |
| hamster | 2294 | house | 2320 | kangaroo | 2563 | keyboard | 1948 |
| lamp | 2242 | lawn_mower | 1929 | leopard | 2139 | lion | 3045 |
| lizard | 2130 | lobster | 2136 | man | 2248 | maple* | 2149 |
| motorcycle | 2168 | mountain | 2249 | mouse | 2128 | mushroom | 2390 |
| oak_tree | 1995 | orange | 2650 | orchid | 1902 | otter | 2073 |
| palm_tree | 2107 | pear | 2120 | pickup_truck | 2478 | pine_tree | 2341 |
| plain | 2198 | plate | 3109 | poppy | 2730 | porcupine | 1900 |
| possum | 2008 | rabbit | 2408 | raccoon | 2587 | ray | 2564 |
| road | 2862 | rocket | 2180 | rose | 2237 | sea | 2122 |
| seal | 2159 | shark | 2157 | shrew | 1826 | skunk | 2450 |
| skyscraper | 2298 | snail | 2369 | snake | 2989 | spider | 3024 |
| squirrel | 2374 | streetcar | 1905 | sunflower | 2761 | sweet_pepper | 1983 |
| table | 3137 | tank | 1897 | telephone | 1889 | television | 2973 |
| tiger | 2603 | tractor | 1848 | train | 3020 | trout | 2726 |
| tulip | 2160 | turtle | 2438 | wardrobe | 2029 | whale | 2597 |
| willow_tree | 2040 | wolf | 2423 | woman | 2446 | worm | 2945 |

