# Peer review of "Temporal Ensembling for Semi-Supervised Learning"

_ICLR 2017 — accepted_

[Public Comment · (anonymous) · 08 Dec 2016]
**Out of curiosity: did you compute MNIST res?**

Out of curiosity: did you compute MNIST results? I was just wondering, since the Ladder paper presented those as well.

[Author Response · Samuli Laine · 14 Dec 2016]
**Revision**

It has come to our attention that a recent paper titled "Regularization With Stochastic Transformations and Perturbations for Deep Semi-Supervised Learning" by Sajjadi et al., presented at NIPS 2016, builds on the same core principle as our work. We have therefore uploaded a new revision of our paper to cite this related work and to contrast our contributions against it. We also added discussion about our current understanding of the reasons why our methods help in the corrupted labels test. To address the points raised by Reviewer #2, we clarified that the main reason for not using SVHN extra data was to be comparable to the previous best results in semi-supervised learning, bolded the best-in-category results in Tables 1 and 2, and added a bit of additional detail about function w(t) where it is first introduced.
 
We look forward to augmenting the paper further with results from CIFAR-100 in the near future, as mentioned in our previous response to Reviewer #1.

[Official Review · AnonReviewer2 · rating 8 · confidence 5 · 14 Dec 2016]
**No Title**

This paper presents a semi-supervised technique for “self-ensembling” where the model uses a consensus prediction (computed from previous epochs) as a target to regress to, in addition to the usual supervised learning loss. This has connections to the “dark knowledge” idea, ladder networks work is shown in this paper to be a promising technique for scenarios with few labeled examples (but not only). The paper presents two versions of the idea: one which is computationally expensive (and high variance) in that it needs two passes through the same example at a given step, and a temporal ensembling method that is stabler, cheaper computationally but more memory hungry and requires an extra hyper-parameter. 


My thoughts on this work are mostly positive. The drawbacks that I see are that the temporal ensembling work requires potentially a lot of memory, and non-trivial infrastructure / book-keeping for imagenet-sized experiments. I am quite confused by the Figure 2 / Section 3.4 experiments about tolerance to noisy labels: it’s *very* incredible to me that by making 90% of the labels random one can still train a classifier that is either 30% accurate or ~78% accurate (depending on whether or not temporal ensembling was used). I don’t see how that can happen, basically.


Minor stuff:
Please bold the best-in-category results in your tables. 
I think it would be nice to talk about the ramp-up of w(t) in the main paper. 
The authors should consider putting the state of the art results for the fully-supervised case in their tables, instead of just their own.
I am confused as to why the authors chose not to use more SVHN examples. The stated reason that it’d be “too easy” seems a bit contrived: if they used all examples it would also make it easy to compare to previous work.

[Official Review · AnonReviewer3 · rating 9 · confidence 4 · 16 Dec 2016]

This work explores taking advantage of the stochasticity of neural network outputs under randomized augmentation and regularization techniques to provide targets for unlabeled data in a semi-supervised setting. This is accomplished by either applying stochastic augmentation and regularization on a single image multiple times per epoch and encouraging the outputs to be similar (Π-model) or by keeping a weighted average of past epoch outputs and penalizing deviations of current network outputs from this running mean (temporal ensembling). The core argument is that these approaches produce ensemble predictions which are likely more accurate than the current network and are thus good targets for unlabeled data. Both approaches seem to work quite well on semi-supervised tasks and some results show that they are almost unbelievably robust to label noise.

The paper is clearly written and provides sufficient details to reproduce these results in addition to providing a public code base. The core idea of the paper is quite interesting and seems to result in higher semi-supervised accuracy than prior work. I also found the attention to and discussion of the effect of different choices of data augmentation to be useful.	

I am a little surprised that a standard supervised network can achieve 30% accuracy on SVHN given 90% random training labels. This would only give 19% correctly labeled data (9% by chance + 10% unaltered). I suppose the other 81% would not provide a consistent training signal such that it is possible, but it does seem quite unintuitive. I tried to look through the github for this experiment but it does not seem to be included. 

As for the resistance of Π-model and temporal ensembling to this label noise, I find that somewhat more believable given the large weights placed on the consistency constraint for this task. The authors should really include discussion of w(t) in the main paper. Especially because the tremendous difference in w_max in the incorrect label tolerance experiment (10x for Π-model and 100x for temporal ensembling from the standard setting).

Could the authors comment towards the scalability for larger problems? For ImageNet, you would need to store around 4.8 gigs for the temporal ensembling method or spend 2x as long training with Π-model.

Can the authors discuss sensitivity of this approach to the amount and location of dropout layers in the architecture? 

Preliminary rating:
I think this is a very interesting paper with quality results and clear presentation. 

Minor note:
2nd paragraph of page one 'without neither' -> 'without either'

[Official Review · AnonReviewer1 · rating 7 · confidence 4 · 18 Dec 2016]
**simple approach showing some decent results**

This paper presents a model for semi-supervised learning by encouraging feature invariance to stochastic perturbations of the network and/or inputs.  Two models are described:  One where an invariance term is applied between different instantiations of the model/input a single training step, and a second where invariance is applied to features for the same input point across training steps via a cumulative exponential averaging of the features.  These models evaluated using CIFAR-10 and SVHN, finding decent gains of similar amounts in each case.  An additional application is also explored at the end, showing some tolerance to corrupted labels as well.

The authors also discuss recent work by Sajjadi &al that is very similar in spirit, which I think helps corroborate the findings here.

My largest critique is it would have been nice to see applications on larger datasets as well.  CIFAR and SVHN are fairly small test cases, though adequate for demonstration of the idea.  For cases of unlabelled data especially, it would be good to see tests with on the order of 1M+ data samples, with 1K-10K labeled, as this is a common case when labels are missing.

On a similar note, data augmentations are restricted to only translations and (for CIFAR) horizontal flips.  While "standard," as the paper notes, more augmentations would have been interesting to see --- particularly since the model is designed explicitly to take advantage of random sampling.  Some more details might also pop up, such as the one the paper mentions about handling horizontal flips in different ways between the two model variants.  Rather than restrict the system to a particular set of augmentations, I think it would be interesting to push it further, and see how its performance behaves over a larger array of augmentations and (even fewer) numbers of labels.

Overall, this seems like a simple approach that is getting decent results, though I would have liked to see more and larger experiments to get a better sense for its performance characteristics.



Smaller comment: the paper mentions "dark knowledge" a couple times in explaining results, e.g. bottom of p.6.  This is OK for a motivation, but in analyzing the results I think it may be possible to have something more concrete.  For instance, the consistency term encourages feature invariance to the stochastic sampling more strongly than would a classification loss alone.

[Author Response · Samuli Laine · 03 Jan 2017]
**CIFAR-100 and Tiny Images results**

We have uploaded a new revision of the paper that includes additional results from CIFAR-100 and Tiny Images datasets (Section 3.3 + end of Appendix A). Especially the Tiny Images test was quite interesting, and we thank Reviewer #1 for the suggestion. For the sake of consistency, we updated Figure 2 with results from temporal ensembling instead of Pi-model (the difference was small). We also reran the supervised graph for the corrupted label test. This time the supervised results with 80% and 90% corruption were more in line with expectations, indicating a possible error in the original graph or particularly high sensitivity to the randomness in the corruption process.

Code in GitHub has been updated to include all code necessary for recreating the results in the paper.

[Final Decision · Program Chairs · 06 Feb 2017]
**ICLR committee final decision**

The reviewers all agree that this is a strong, well-written paper that should be accepted to the conference. The reviewers would like to see the authors extend the analysis to larger data sets and extend the variety of augmentations. Two of the reviewers seem to suggest that some of the experiments seem too good to be true. Please consider releasing code so that others can reproduce experiments and build on this in future work.